# Estimation of Optimum Vacuum Pressure of Air-Suction Seed-Metering Device of Precision Seeders Using Artificial Neural Network Models



Davut Karayel [1,2,*], Orhan Güngör [3] and Egidijus Šarauskis [2]

1   Department of Agricultural Machinery and Technologies Engineering, Faculty of Agriculture, Akdeniz University, Antalya 07058, Turkey

2   Agriculture Academy, Department of Agricultural Engineering and Safety, Vytautas Magnus University, LT-53362 Kaunas, Lithuania; egidijus.sarauskis@vdu.lt

3   Tefenni Vocational School, Burdur Mehmet Akif Ersoy University, Burdur 15600, Turkey; orhangungor@mehmetakif.edu.tr

\*   Correspondence: dkarayel@akdeniz.edu.tr

**Abstract:** The success of the seed-metering device of a seeder determines the quality seeding and final plant stand. The adjustment of the optimal vacuum pressure of air-suction-type seed-metering devices is a key factor affecting the success of seed-metering devices. The optimal value of vacuum of the seed-metering device should be adjusted in relation to the physical properties of the seed before seeding in the field. This research was carried out to estimate the optimal value of vacuum pressure of an air-suction seed-metering device of a precision seeder by using an artificial neural network method. Training of the network was performed by using a Levenberg–Marquardt (LM) learning algorithm. Training and testing were carried out using Matlab software. The inputs were physical properties of seeds such as surface area, thousand kernel weight, kernel density and sphericity. Optimum vacuum pressures were determined for soybean, maize, cucumber, melon, watermelon, sugarbeet and onion seeds in laboratory. Surface area, thousand kernel weight, kernel density and sphericity of seeds varied from 0.05 to 0.638 cm$^2$, 4.4 to 322.4 g, 0.43 to 1.29 g cm$^{-3}$ and 42.8 to 85.75%, respectively. The optimal vacuum pressure was determined as 1.5 kPa for onion; 2.0 kPa for sugarbeet; 2.5 kPa for melon and watermelon; 3.0 kPa for soybean; and 4.0 kPa for maize seeds. A trained program using an artificial neural network could satisfactorily estimate the optimum value of vacuum pressure of the air-suction type seed-metering device of precision seeders with a prediction success (R$^2$) of 0.9949 for both linear and polynomial regressions.

**Keywords:** artificial neural networks; vacuum seeder; seed metering; seed distribution uniformity; precision seeding

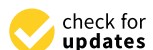

## 1. Introduction

Seeding is the process of putting seeds at a certain depth of the soil with a suitable distribution for plant requirements and closing them for plant production. Seeding methods can generally be grouped into three groups according to seed distribution over field: broadcasting, banding and drilling. While the seeds are scattered to 100% of the field surface in the broadcasting method, the seeds are scattered to 50% and 10% of the field surface in banding and drilling, respectively. In precision seeding, the seeds are sown in furrows, and the spacing of seeds within the furrows is uniform. Transplanting seedlings into a field is the fourth method of planting. Seeders or transplanters have been developed or manufactured to enable each of these seeding or transplanting methods [1,2].

A precision seeder is a type of row-crop seeder that is designed to deposit a single seed at equal row intervals. When precise seeding is considered, the seeding of plants such as sugarbeet, corn, cotton and soybean is the first thought. In addition, the precision seeding method is not yet widely used in vegetable production, which has a higher economic value.

The precision seeders can be classified according to seed-metering systems. While there is a large range of precision metering systems, most can be broadly classified as finger pick-up, plate (horizontal, inclined, and vertical plate types), belt, air-suction (vacuum disc) and pressurized drum types. The classification of metering systems largely depends on the design of the seed singulation (selection of single seeds from seed hopper) part of metering systems that enables seed singulation [3].

The most used metering device in precision seeders are air-suction-type precision metering devices. This metering devices consist of a vertical rotating disc that has a row of holes around its circumference, a seed hopper, and a fan or blower. The vertical disc differs from the plate used in plate-type precision metering systems in that the seeds do not fall into, nor pass through, the hole. The hole diameter of vertical disc should be smaller than the smallest cross-sectional dimension of the seed in the seed lot.

In many studies carried out with precision seeders equipped with an air-suction-type metering device, the success of the seeder has been emphasized for some seeds such as cotton, maize, soybean, sugarbeet, rape and onion [4–9]. Xu et al., determined structural parameters of air-suction type seed-metering device using the DEM-CFD coupling method. The optimal combination of seeding performance parameters in the air-suction seed-metering device were a seed-throwing angle of 13° and negative pressure of 3.1 kPa for pelletized vegetable seeds. When the optimal combination of parameters adjusted on seeder, the quality of feed index, miss and multiple indexes of seed-metering device were 95.9, 2.9 and 1.2%, respectively [10]. The main parameters affecting the seeding performance of air-suction-type metering devices are vacuum pressure, angular velocity of the metering tray, and taper angle of the sucking hole. Sun et al., identified the optimum level of these parameters as a vacuum pressure of 2.16 kPa, angular velocity of 29.43 rpm and taper angle of 61.51° for Chinese cabbage [11]. Karayel et al., analyzed the relationship between physical property of the seeds and vacuum pressure of an air-suction-type seed-metering device and determined the optimum vacuum pressure of a precision vacuum seeder by developing mathematical models using the physical properties of seeds [8].

Various methods such as grease belt, optical sensor or high-speed camera are used to evaluate the seeding performance of the seeders in laboratory. Among these methods, the most commonly used method is a grease belt. Although it is a convenient method for seeders, there are some restrictions. The greased belt length limits the data that can be taken, and it is time consuming. Nevertheless, it has been used by many researchers to determine the seeding quality of seeders. Easier and faster methods are needed to determine or estimate the design and operating parameters (such as vacuum pressure and feed rate) of seeders [12]. Artificial neural network (ANN) models can be an alternative to field or laboratory experiments for the determination of the optimum values of design and operating parameters of seeders.

Artificial neural network is a research area under the science of artificial intelligence in which researchers are very interested. It concerns the study of computer learning. Artificial intelligence technology is developing at an increasing rate. New products are emerging that apply artificial neural networks to everyday life. Automation systems are equipped with artificial intelligence technology to take advantage of the decision-making power of the computer. Increasing numbers of commercial systems are emerging each day, and the functional properties of the systems are increasing.

Artificial intelligence technologies include:

- Expert systems;
- Artificial neural networks;
- Genetic algorithms;
- Fuzzy logic; and
- Hybrid systems [13].

These technologies contribute to the formation of useful products for people in daily life. Artificial neural networks provide computer learning. Machine learning is defined as

the improvement of behavior over time. Different learning paradigms have been developed, and all these learning paradigms are based on three strategies. These are:

- Supervised learning;
- Unsupervised learning; and
- Reinforcement learning [14].

Based on these strategies, there are rules of learning development. Some of these rules are online and others are offline learning.

Scientific research on the modeling of the performance or design parameters of agricultural machinery using artificial neural network are very limited. Anantachar et al., developed feed-forward artificial neural network (ANN) models for the prediction of performance parameters (seed rate, seed spacing and percent seed damage) of an inclined disc seed-metering device. Forward speed, peripheral speed of the metering mechanism and area of the cells on the seed disc were input parameters for artificial neural network (ANN) method. The results show that the ANN model predicted the performance parameters of the seed-metering mechanism better than statistical methods. It was observed that the optimum forward speed of the seeder and optimum area of cells on the seed-metering disc had good correlation with seed size. Optimum forward speed of seeder and optimum size (area) of slots on the seed-metering plate were estimated using developed regression equations. The size range of the peanut seeds used was between 95.42 and 123.01 mm$^2$, and the optimum peripheral speed of seed-metering system was 0.237 m/s for these seeds. The authors recommended that the results need to be verified by conducting field experiments [15].

Zhao et al., predicted the performance of a precision seeder using artificial neural network and grey model. Seed motion in soil was simulated using rectangular vibrating tray applying discrete element method. According to results of the research, the proposed method had good precision and stability to promote a uniform seed distribution [16].

The performance of a seeder and seeding quality depend on seeder design, seed quality, climate conditions, soil conditions, proper adjustment of the seeder for used seeds and skill of the operator. Storage, cleaning, sorting, separation and seeding equipment in various types are designed by considering the physical characteristics of the seeds. These characteristics should be considered not only in the design of these equipment but also in the determination of the operating parameters.

In this research, using some of these properties, e.g., surface area, thousand seed weight, seed density and sphericity, soft computing-based models such as artificial neural network have been developed to estimate the optimum vacuum pressure. The Levenberg–Marquardt (LM) learning algorithm was chosen as the learning algorithm of the model. The LM is the most preferred algorithm in the medicine classification, manufacturing industry and engineering [17,18]. The most important reason for the preference for the LM algorithm is its speed and stability in artificial neural network training.

The LM algorithm formula derived from steepest descent and Newton algorithms is given in Equation (1). The Newton algorithm supports the speed, while the Steepest descent algorithm supports the LM algorithm [19].

$$\Delta w = \left( J^T J + \mu I \right)^{-1} J^T e \tag{1}$$

where $w$ is the weight vector, $J$ is the Jacobian matrix, $\mu$ is the combination coefficient, $I$ is the unit matrix, and $e$ is the cumulative error vector. If this parameter is too large, the method behaves like the Newton method; if it is too small, it behaves as the steepest descent method.

## 2. Materials and Methods

A Sönmezler model PMD (Sönmezler, Adana, Turkey) precision seeder, equipped with the air-suction type metering device, was used in all experiments. The laboratory tests were performed to determine the seed-metering uniformity of the metering device under different vacuum pressure regulations with the different seeds: soybean, maize,

cucumber, melon, watermelon, sugarbeet and onion. All seeds were uncoated (bare) seed. The seed-metering device was adjusted to longitudinal seed spacings as close as possible to the seed spacing recommended by the seed producer. These seeds represent several shapes, varying from spherical to flat and elongated. Length, width, and thickness of the seeds are presented in Table 1.

**Table 1.** Dimensions of seeds.

| Seed | Length, mm | Width, mm | Thickness, mm |
|---|---|---|---|
| Soybean | 7.93 ± 0.12 | 6.75 ± 0.07 | 5.78 ± 0.07 |
| Maize | 10.65 ± 0.21 | 7.98 ± 0.12 | 4.53 ± 0.06 |
| Cucumber | 10.21 ± 0.18 | 3.98 ± 0.08 | 1.63 ± 0.02 |
| Melon | 11.58 ± 0.17 | 4.51 ± 0.07 | 2.08 ± 0.03 |
| Watermelon | 8.9 ± 0.14 | 6.33 ± 0.08 | 2.05 ± 0.03 |
| Sugarbeet | 4.55 ± 0.06 | 3.58 ± 0.07 | 2.56 ± 0.05 |
| Onion | 2.25 ± 0.03 | 1.51 ± 0.04 | 1.33 ± 0.03 |

The metering device of the seeder used the vertical seed disc (plate) for singulating the seed from the seed hopper (Figure 1). Air suction (vacuum) caused the seed to stick to the holes of seed disc. The stuck seed rotated with the vertical seed disc, and it was released from the disc by airflow cut-off at the bottom of the seed disc. The absence of vacuum (air suction) allowed the seed to be dropped into soil. The diameters of the seed discs were 0.23 m. The holes on seed disc were drilled along a circle of 0.2 m diameter. The diameter of holes on the seed disc were 1.5 mm for sugarbeet and onion; 2.5 mm for melon, watermelon and cucumber; and 3.5 mm for maize and soybean. The peripheral speed of the metering disc was 0.25 m s$^{-1}$.

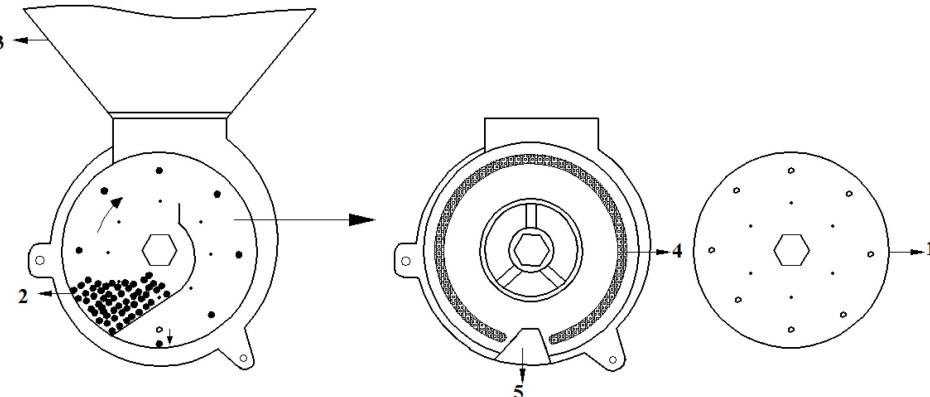

**Figure 1.** The air-suction-type seed-metering devices: 1, vacuum plate; 2, seed; 3, seed box; 4, air suction canal; 5, air cut.

The optimum value of vacuum pressure for each seed was determined primarily by experiments in the laboratory. Then, it was estimated by using an artificial neural network method using the physical properties of the seeds (surface area, thousand kernel weight, kernel density and sphericity). Finally, optimum vacuum values obtained by both methods were compared.

Seed distribution uniformity of the metering device of precision seeder was determined using a grease belt test stand in laboratory. The dimensions of the belt of test stand were 0.15 m × 7.5 m. The precision seeder was mounted on a seeder test stand. An adjustable speed drive mechanism was utilized to operate the seed-metering device of the seeder. An adequate amount of grease was applied to the top surface of the belt to capture the seed as it was dropped from the seeder to avoid rolling or bouncing of the seed on the belt surface. The seeder was tested with a forward speed of 1.5 m s$^{-1}$. The vacuum pressure of the seed-metering unit was adjusted for the values between 2 and 5 kPa for

soybean and maize, 2 and 3.5 kPa for melon, watermelon and cucumber and 1 and 2.5 kPa for sugarbeet and onion seeds.

Longitudinal spacings between successive seeds were measured along the 6 m length of the greased belt. The number of holes on seed disc and transmission ratio of the seed-metering device were adjusted to ensure a theoretical seed spacing of 110 mm for soybean, 225 mm for maize, 580 mm for melon, watermelon and cucumber, 160 mm for sugarbeet and 80 mm for onion.

Researchers (or engineers) use a wide variety of measurement procedures to qualify the sowing performance (seed distribution uniformity) of seeders in relation to seed or plant spacing [12,20,21]. Some researchers use performance measures involving successive seed spacings on grease belt test stand or by electronic measurement systems including optic sensors or image processing procedures [22–24]. Other researchers use performance measures involving successive plant spacings in the field. A few researchers have used performance measures involving distance between successive seeds sown into furrow [9,21,25].

Kachman and Smith calculated multiple index (MULTI), miss index (MISS), quality of feed index (QFI) and preciseness (PREC) from seed spacings for analyzing the sowing uniformity of precision seeders [26]. MISS is the ratio of seed spacings higher than 1.5 times the adjusted seed spacing and indicates the percentage of missed seed standings or skips. MULTI is the ratio of seed spacings that are less than or equal to half of the adjusted seed spacing and indicates the percentage of multiple seed drops. QFI is the ratio of spacings between 0.5 and 1.5 times the adjusted seed spacing. QFI is 100% minus miss and multiple indices and indicates the percentages of single seed drops or locations. PREC is the coefficient of variation of the spacings that are classified as singles after neglecting the spacings consisting of misses and multiples.

The physical properties of the seeds were measured using the following methods:

A vernier caliper with a sensitivity of 0.01 mm was used to measure the thickness, width and length of the seeds. The sphericities ($\phi$) of the seeds were calculated with Equation (2) [27]:

$$\phi = \frac{(L \times W \times T)^{\frac{1}{3}}}{L} \times 100 \qquad (2)$$

where $L$ is the length of the seeds, $W$ is the width of the seeds, and $T$ is the thickness of the seeds in mm.

The liquid displacement method was used to measure the kernel density of seeds. Since toluene ($C_7H_8$) is not absorbed by the seeds, it was used rather than water [27]. A digital camera and Adobe Photoshop software were used to measure the surface area of seeds [8,28].

Feed-forward artificial neural networks were used to estimate the optimal vacuum pressure according to the physical properties of seeds. The suitability of the artificial neural network was compared and evaluated using the coefficient of determination (R squared) of regression curves. Coefficient of determination is the proportion of variance in the dependent variable (experimental vacuum pressure) that is predictable from the independent variable (predicted vacuum pressure). The higher values of the coefficient of determination, the better the goodness of the fit, and its highest value is 1.

In the present work, a neural network with the architecture of a back propagation (backprop, BP) learning algorithm based on the Levenberg–Marquardt algorithm (LM) was developed to predict the optimum vacuum pressure of an air-suction-type precision metering device using the MATLAB Neural Network Toolbox. The backpropagation algorithm requires the definition of a network structure consisting of one or more layers, where one layer is completely dependent on the next layer. A standard network structure consists of an input layer, a hidden layer, and an output layer. The program used the physical properties of seeds such as thousand kernel weight, surface area, sphericity and kernel density for inputs. Vacuum pressure value was obtained as the output. Hidden layer neurons are directly connected to the input layer before it and the output layer after it. Four neurons (thousand kernel weight, surface area, sphericity and kernel density)

were considered in the input layer and one neuron in the output layer to represent the input parameters and the response, respectively. The appropriate number of neurons in the hidden layers was chosen by trial-and-error method [29]. Five hidden layers were considered for the better approximation of the output parameters. While 70% of the samples were used for trainings, 15% were used for validation and 15% for testing. The representation of the ANN is shown in Figure 2.

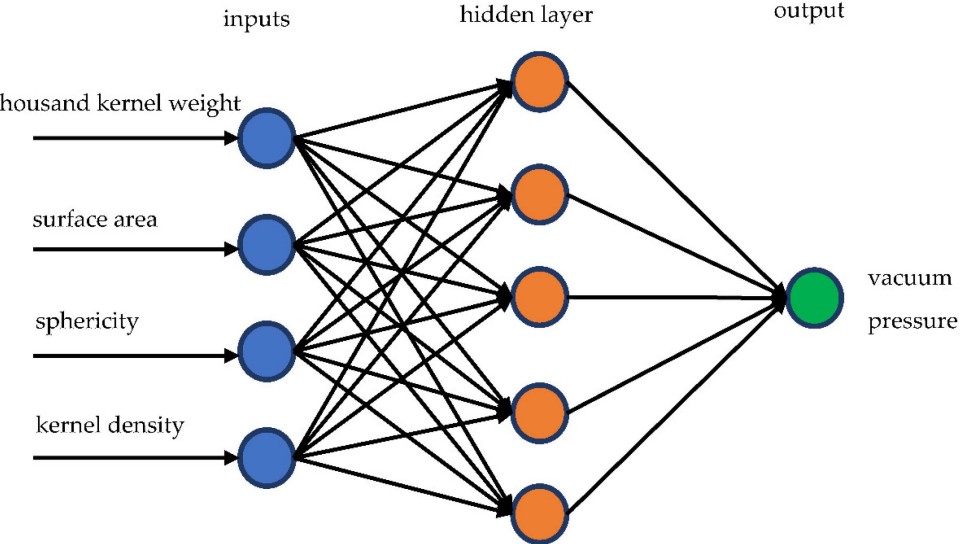

**Figure 2.** ANN representation used in vacuum pressure estimation.

Each experiment was replicated six times in a laboratory. ANOVA (analysis of variance) was performed to analyze the data sets. Duncan's multiple-range tests were used to identify significantly different means within dependent variables.

## 3. Results and Discussion

MULTI, MISS, QFI and PREC calculated from seed spacing values of all seeds are given in Tables 2 and 3. The seed distribution uniformity values of the seed-metering device (MULTI, MISS, QFI and PREC) were affected by vacuum pressure. Parameters of QFI and PREC were evaluated to determine the optimal vacuum pressure for each seed. When the QFI is maximum and the PREC is minimum, the vacuum pressure is optimum for each seed. The best seed spacing uniformity with the highest QFI and the lowest PREC was achieved at the vacuum pressure of 1.5 kPa for onion; 2.0 kPa for sugarbeet; 2.5 kPa for cucumber, melon and watermelon; 3.0 kPa for soybean; and 4.0 kPa for maize seeds (Tables 2 and 3).

The most uniform seed distribution uniformity was obtained with soybean seeds at any vacuum pressure level. The uniform and spherical shape of soybean obtained uniform meters of seeds with the air-suction seed-metering system. Increasing the vacuum pressure reduced the MISS and increased MULTI for all seeds. Multiple seeds were more common than misses for cucumber, melon, watermelon, onion and sugarbeet seeds. Few gaps or multiple seed drops happen at any vacuum pressure values for soybean and maize seeds. Our results support reports from Karayel et al., who found that the seed distribution uniformity of precision seeders differed most at lower or higher vacuum pressure values and faster forward speeds of seeders [8]. Because of the higher PREC and lower QFI values, the performance of the seeder (seed distribution uniformity) was poorer at the lower and higher vacuum pressures than at the optimum vacuum pressure.

**Table 2.** Soybean, maize, and cucumber seeds metering uniformity of air-suction seed-metering device of precision seeder.

| Vacuum Pressure, kPa | PREC, % | MULTI, % | MISS, % | QFI, % |
|---|---|---|---|---|
| | | Soybean | | |
| 2.0 | 7.2 | 0.0 b | 8.0 a | 92.0 b |
| 3.0 | 6.1 | 0.1 b | 4.0 b | 95.9 a |
| 4.0 | 6.9 | 3.0 a | 4.0 c | 93.0 b |
| 5.0 | 8.1 | 3.0 a | 4.3 c | 92.7 b |
| | | Maize | | |
| 2.0 | 14.5 | 0.0 c | 9.6 a | 90.4 c |
| 3.0 | 10.8 | 1.1 b | 7.3 a | 91.6 b |
| 4.0 | 8.5 | 1.4 c | 3.7 b | 94.9 a |
| 5.0 | 12.5 | 5.1 a | 2.2 b | 92.7 b |
| | | Cucumber | | |
| 2.0 | 22.7 | 9.8 c | 10.3 a | 79.9 c |
| 2.5 | 15.1 | 7.9 c | 4.1 b | 88.0 a |
| 3.0 | 25.8 | 12.0 b | 5.4 b | 82.6 b |
| 3.5 | 30.9 | 18.1 a | 2.2 b | 79.7 c |

Note: Values followed by the same letter (a, b, c) are not significantly different ($p < 0.05$).

**Table 3.** Melon, watermelon, sugarbeet and onion seeds metering uniformity of air-suction seed-metering device of precision seeder.

| Vacuum Pressure, kPa | PREC, % | MULTI, % | MISS, % | QFI, % |
|---|---|---|---|---|
| | | Melon | | |
| 2.0 | 18.1 | 7.8 b | 11.1 a | 81.1 b |
| 2.5 | 14.1 | 11.5 b | 4.3 b | 84.2 a |
| 3.0 | 22.7 | 17.3 a | 3.8 b | 78.9 b |
| 3.5 | 26.2 | 21.9 a | 1.8 c | 76.3 b |
| | | Watermelon | | |
| 2.0 | 27.8 | 6.9 c | 12.9 a | 80.2 c |
| 2.5 | 15.1 | 9.5 c | 1.9 b | 88.6 a |
| 3.0 | 18.9 | 14.3 b | 2.1 b | 83.6 b |
| 3.5 | 28.9 | 15.1 a | 1.9 b | 83.0 b |
| | | Sugarbeet | | |
| 1.0 | 33.5 | 12.8 b | 10.1 a | 77.1 c |
| 1.5 | 31.3 | 11.9 b | 7.4 b | 80.7 b |
| 2.0 | 19.8 | 11.1 b | 3.5 c | 85.4 a |
| 2.5 | 41.9 | 17.9 a | 2.8 c | 79.3 c |
| | | Onion | | |
| 1.0 | 41.3 | 11.8 c | 4.7 a | 83.5 b |
| 1.5 | 23.9 | 12.9 c | 0.4 b | 86.7 a |
| 2.0 | 30.9 | 24.9 b | 0.1 b | 75.0 c |
| 2.5 | 44.2 | 32.6 a | 0.0 b | 67.4 d |

Note: Values followed by the same letter (a, b, c, d) are not significantly different ($p < 0.05$).

Physical properties of the seeds as presented in Table 4 were used as input parameters in the program trained using artificial neural network. The thousand kernel weight, surface area, sphericity and kernel density of seeds varied from 4.4 to 322.4 g, 0.05 to 0.638 cm², 42.8 to 85.75% and 0.43 to 1.29 g cm$^{-3}$, respectively. The thousand kernel weight, surface area, sphericity and kernel density were used as data inputs for training the program, and as a result, optimum vacuum pressure values were predicted (Table 5). The learning rate was used to change the weights of the ANN. If the learning rate is low, then training is more reliable, but optimization will take a lot of time because steps towards the minimum

of the loss function are tiny. If the learning rate is high, then training may not converge or even diverge. Narrow areas where learning will take place can be skipped. The results are better by using the learning rate in the range of $0.01 \leq \eta \leq 0.9$. A learning rate of 0.3 was found to be adequate for this network.

**Table 4.** Means and standard errors of the seed dimensions.

| Seed | Thousand Kernel Weight, g | Surface Area, cm$^2$ | Sphericity, % | Kernel Density, g cm$^{-3}$ |
|------|---------------------------|----------------------|---------------|-----------------------------|
| Soybean | $206.50 \pm 1.12$ | $0.450 \pm 0.005$ | $85.75 \pm 0.47$ | $1.130 \pm 0.012$ |
| Maize | $322.40 \pm 2.0$ | $0.638 \pm 0.005$ | $81.80 \pm 0.54$ | $1.280 \pm 0.011$ |
| Cucumber | $29.20 \pm 0.17$ | $0.270 \pm 0.003$ | $36.90 \pm 0.26$ | $1.080 \pm 0.011$ |
| Melon | $35.80 \pm 0.17$ | $0.360 \pm 0.003$ | $42.80 \pm 0.28$ | $0.430 \pm 0.005$ |
| Watermelon | $45.10 \pm 0.27$ | $0.310 \pm 0.003$ | $59.80 \pm 0.40$ | $1.290 \pm 0.011$ |
| Sugarbeet | $13.70 \pm 0.14$ | $0.090 \pm 0.001$ | $71.90 \pm 0.42$ | $0.920 \pm 0.008$ |
| Onion | $4.40 \pm 0.04$ | $0.050 \pm 0.001$ | $71.20 \pm 0.46$ | $1.050 \pm 0.010$ |

**Table 5.** Experimental and predicted (using artificial neural network) optimum vacuum pressure values.

| Seed | Experimental Vacuum Pressure, kPa | Predicted Vacuum Pressure, kPa |
|------|-----------------------------------|--------------------------------|
| Soybean | 3.0 | 2.997 |
| Maize | 4.0 | 3.997 |
| Cucumber | 2.5 | 2.419 |
| Melon | 2.5 | 2.498 |
| Watermelon | 2.5 | 2.499 |
| Sugarbeet | 2.0 | 1.840 |
| Onion | 1.5 | 1.493 |

The validation of the predicted vacuum pressures (the performance of ANN) was evaluated by comparing the predicted and measured (experimental) vacuum pressures (Figure 3). Both polynomial and linear regression curves showing the accuracy of the vacuum values obtained by ANN and success rates (coefficient of determination) are shown in the figures.

The performance of the ANN model (with the configuration of backpropagation) for the prediction of the optimum vacuum pressure of the seed-metering device of a precision seeder is quite good within the given range of independent parameters, because the predicted values of vacuum pressure generally banded around a straight line. As confirmed by both regression curves, the predictive success is over 99%. The results show that the ANN model estimate the optimal vacuum pressure of an air-suction-type seed-metering device better than the mathematical models (statistical regression models) presented in Karayel et al. [8]. Our results support reports from Anantachar et al., who found the prediction accuracy of ANN models for the performance of a seed-metering device (percent seed damage, seed spacing and seed rate) better than that of statistical models developed using regression analysis [15], because ANN models have the ability to entirely capture the input–output dataset correlation during training and have a better generalization performance and flexibility.

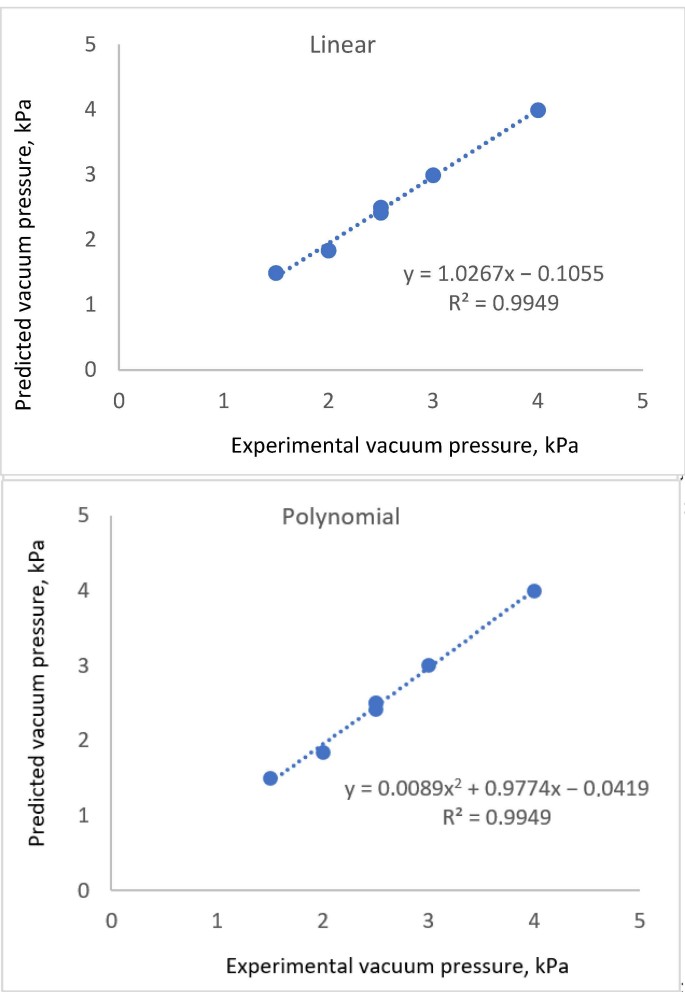

**Figure 3.** Polynomial and linear regression curves.

### 4. Conclusions

The level of vacuum pressure is one of the most important factors affecting the seed singulation uniformity of precision vacuum metering devices and must be adjusted precisely before starting the seeding operation. The physical properties of the seeds should be considered in the selection of its appropriate level.

An artificial neural network was developed to estimate the optimum vacuum pressure of an air-suction seed-metering device of a precision seeder. The program used some physical properties of seeds (surface area, thousand seed weight, seed density and sphericity) as inputs. The Levenberg–Marquardt (LM) learning algorithm was chosen as the learning algorithm of the model. The performance of the neural network model was compared with laboratory tests.

In laboratory experiments of an air-suction seed-metering device, the optimum value of vacuum pressure was determined as 3.0 kPa for soybean; 4.0 kPa for maize; 2.5 kPa for melon and watermelon; 2.0 kPa for sugarbeet; and 1.5 kPa for onion seeds. The artificial neural network was a sufficiently satisfactory method for predicting the optimum vacuum pressure of the seed-metering device of seeders, with a prediction success over 0.99.

**Author Contributions:** Conceptualization, methodology, validation, investigation, writing—original draft preparation, D.K.; conceptualization, methodology, validation, software, O.G.; conceptualization, methodology, investigation, writing—original draft preparation writing, E.Š. All authors have read and agreed to the published version of the manuscript.

**Funding:** This research received no external funding.

**Data Availability Statement:** Not applicable.

**Conflicts of Interest:** The authors declare no conflict of interest.

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
