# Peer review of "Estimation of Optimum Vacuum Pressure of Air-Suction Seed-Metering Device of Precision Seeders Using Artificial Neural Network Models"

_agronomy, doi:10.3390/agronomy12071600_

Round 1

Reviewer 1 Report

This manuscript will be interesting for scientists, who develop and research the seed-metering device of a seeder. Also, these scientific results will have practical application. In order to improve the manuscript, I suggest the following corrections:

1. The article states: “While the seeds are scattered to 100% of the field sur- 37 face in the broadcasting method, the seeds are scattered to 50% and 10% of the field sur- 38 face in banding and drilling, respectively” (p. 1, lines 37-39). It is necessary to indicate the source where this information can be verified.

2. If the classification of precision seed metering systems is not proposed by the authors, it is necessary to indicate the source from which such information is taken (p. 2, lines 48-50).

3. The design of air-suction type precision metering devices is described in the article (p. 2, lines 54-58). It is advisable to present the scheme of precision metering devices in the article.

4. Methods for evaluating the seeding performance of seeders in the laboratory are mentioned in the article (p. 2, lines 68-72). It is necessary to indicate the source where this information can be verified or where these methods are described.

5. The components of Artificial intelligence technologies are presented in the article (p. 2, lines 83-88). If it is not proposed by the authors, it is necessary to indicate the source from which such information is taken.

6. The article states: “Different learning paradigms have been developed and all these learning paradigms are based on three strategies” (p. 2, lines 91-95). These strategies are listed in the article, but the source of this information is missing.

7. The precision seeder, equipped with the air-suction type metering device, was used in all experiments (p. 3, lines 139-140). The model of the precision seeder must be indicated.

8. The rotation speed of the vertical seed disc of the metering device must be specified (p. 4, lines 148-151).

9. The method of measuring the surface area of seeds should be described in more detail or the resource where it can be found should be indicated.

10. The article states: “Display output of artificial neural network prediction method (data input and output 273 cells) is presented in Figure 3” (p. 7, lines 273-274). The article does not contain Figure 3.

11. The unit (kPa) of Experimental vacuum pressure and Predicted vacuum pressur should be marked on the axes of Figure 2.

Author Response

Reviewer 1

Thank you very much for your valuable comments. Your comments are quite useful to improve the paper. We have improved the manuscript and made further changes accordingly. We hope these changes will meet with your approval. These changes are listed below.

  1. The article states: “While the seeds are scattered to 100% of the field sur- 37 face in the broadcasting method, the seeds are scattered to 50% and 10% of the field sur- 38 face in banding and drilling, respectively” (p. 1, lines 37-39). It is necessary to indicate the source where this information can be verified.

Response: The reference for this information has been added (line 42)

  1. If the classification of precision seed metering systems is not proposed by the authors, it is necessary to indicate the source from which such information is taken (p. 2, lines 48-50).

Response: The source reference for classification of precision metering systems has been added (line 52)

  1. The design of air-suction type precision metering devices is described in the article (p. 2, lines 54-58). It is advisable to present the scheme of precision metering devices in the article.

Response: The scheme of precision metering devices has been added (page 4 - Figure 1)

  1. Methods for evaluating the seeding performance of seeders in the laboratory are mentioned in the article (p. 2, lines 68-72). It is necessary to indicate the source where this information can be verified or where these methods are described.

Response: The reference for this information has been added (line 82)

  1. The components of Artificial intelligence technologies are presented in the article (p. 2, lines 83-88). If it is not proposed by the authors, it is necessary to indicate the source from which such information is taken.

Response: The source reference of this information has been added (line 97)

  1. The article states: “Different learning paradigms have been developed and all these learning paradigms are based on three strategies” (p. 2, lines 91-95). These strategies are listed in the article, but the source of this information is missing.

Response: The source reference of this information has been added (line 104)

  1. The precision seeder, equipped with the air-suction type metering device, was used in all experiments (p. 3, lines 139-140). The model of the precision seeder must be indicated.

Response: The model of the precision seeder has been indicated in line 150

  1. The rotation speed of the vertical seed disc of the metering device must be specified (p. 4, lines 148-151).

Response: The rotation speed of the seed disc of the metering device has been presented in line 167-168.

  1. The method of measuring the surface area of seeds should be described in more detail or the resource where it can be found should be indicated.

Response: The reference for this information has been added (line 220 and 221)

  1. The article states: “Display output of artificial neural network prediction method (data input and output 273 cells) is presented in Figure 3” (p. 7, lines 273-274). The article does not contain Figure 3.

Response: Thanks for the reviews' advice. We are sorry for making the mistake. The lines (between 273-274 of previous version) has been removed.

  1. The unit (kPa) of Experimental vacuum pressure and predicted vacuum pressure should be marked on the axes of Figure 2.

Response: The unit (kPa) of experimental vacuum pressure and predicted vacuum pressure has been marked on the axes of Figure 3

Reviewer 2 Report

I regret to state that the manuscript is not well written, missing not only the clearness of the language but also too much information necessary to be able to reproduce the presented results, thus not deserving publication in Agronomy. 

1.         The background literature is less relevant to the research content.

2.         The author lacks the preliminary investigation of the content of this research, and domestic and foreign scholars have carried out a lot of research on vacuum pressure related to the physical properties of the seeds. Hence the introduction part, ‘there are limited study on determination of optimal operational parameters of seeders such as vacuum pressure related to the physical properties of the seeds’ is wrong.

3.         The training data is too small, so there is no statistical significance in predicting vacuum pressure based on the physical properties of the seeds using ANN.

4.         The physical property of the seed is a range value, and the vacuum pressure is a range value, so it remains to be demonstrated whether the relationship between the two can be quantitatively analyzed.

5.         The optimal vacuum pressure of different seeds has been determined experimentally. Why should predict again? The measured value is definitely more meaningful than the predicted value.

6.         The prediction accuracy of linear and multiple regression methods is already very accurate, why use ANN to model this again, except to complicate the problem, it has no sense.

7.         Not only is the detailed description of the ANN modeling stage lacking, but also the description of the ANN modeling results.

Author Response

Reviewer 2

Thank you very much for your valuable comments. Your comments are quite useful to improve the paper. We have improved the manuscript and made further changes accordingly. We hope these changes will meet with your approval. These changes are listed below.

  1. The background literature is less relevant to the research content.

Response:

This research was carried out to determine an easier and faster methods to determine the operating parameters (such as vacuum pressure) of a seed-metering device of seeder and optimisation of performance of seeder. We think that artificial neural network (ANN) models can be an alternative to field or laboratory experiments for the determination of the optimum values of design or operating parameters of seeders. The research content includes measurement methods of sowing quality of seeders, optimisation of performance of seed metering device of seeder and applying the ANN models for modelling of the performance parameters of seeders. Therefore, the background literature of manuscript is related with the measurement methods applied to determine sowing performance (seed distribution uniformity) of seeders, the optimisation of seed metering devices [References 4-11] and applying the ANN models for modelling of the performance parameters of seeders [References 15-16]. We have added new literatures in Introduction considering the recommendation of the reviewer (lines 67-74). We believe that the literatures are relevant to research content.

  1. The author lacks the preliminary investigation of the content of this research, and domestic and foreign scholars have carried out a lot of research on vacuum pressure related to the physical properties of the seeds. Hence the introduction part, ‘there are limited study on determination of optimal operational parameters of seeders such as vacuum pressure related to the physical properties of the seeds’ is wrong.

Response: The introduction part, ‘there are limited study on determination of optimal operational parameters of seeders such as vacuum pressure related to the physical properties of the seeds’ have removed from manuscript.

  1. The training data is too small, so there is no statistical significance in predicting vacuum pressure based on the physical properties of the seeds using ANN.

Response: We used seven different types of seed. We selected the seeds with different physical properties, especially different sizes. The number of seeds may be low, but the variation width in the physical properties of seeds (seed size) has been tried to be as high as possible. The selected seeds for experiments represent several shapes varying from spherical to flat and elongated.

  1. The physical property of the seed is a range value, and the vacuum pressure is a range value, so it remains to be demonstrated whether the relationship between the two can be quantitatively analysed.

Response: The relationship between physical property of the seed and the vacuum pressure was quantitavely analyzed and presented by Karayel et al 2004 (cited in this manuscript). This stuation has been mentioned in lines 71-74.

  1. The optimal vacuum pressure of different seeds has been determined experimentally. Why should predict again? The measured value is definitely more meaningful than the predicted value.

Response: We want to develop easier and faster methods using ANN instead of laboratory and field experiments. This situation has been mentioned in lines 80-84.

  1. The prediction accuracy of linear and multiple regression methods is already very accurate, why use ANN to model this again, except to complicate the problem, it has no sense. The prediction accuracy of linear and multiple regression methods is already very accurate

Response: The linear and polynomial models are not used for prediction of vacuum pressure. These models were used for validation of the predicted vacuum pressure using the ANN model by comparing the measured (experimental) vacuum pressures.

Both polynomial and linear regression curves are showing the accuracy of the vacuum values obtained by ANN compared with measured vacuum pressures (lines between 223-228 and 291-295)

  1. Not only is the detailed description of the ANN modelling stage lacking, but also the description of the ANN modelling results.

Response: Some explanations on ANN modelling stage (lines 229-243) and ANN modelling results (lines 284-290) have been added.

Reviewer 3 Report

The paper presents a theoretical scientific approach, does not make concrete references to improve the performance of the working process of precision seeders. Therefore, I recommend the authors to use the results obtained to make concrete and relevant proposals to increase the sowing accuracy of machines in this category.  

Author Response

Reviewer 3

Thank you very much for your valuable comment. Your comment is quite useful to improve the paper. We have improved the manuscript and made further changes accordingly. We hope this change will meet with your approval. This change is explained below.

The paper presents a theoretical scientific approach, does not make concrete references to improve the performance of the working process of precision seeders. Therefore, I recommend the authors to use the results obtained to make concrete and relevant proposals to increase the sowing accuracy of machines in this category.

Response: Some relevant proposals to increase sowing accuracy of seeders regarding the results of this research have been added to Conclusions (lines 319-322) and some other proposals were also present in previous and new_revised version of manuscript (lines 331-333).

Round 2

Reviewer 3 Report

The analyzed paper presents toretic aspects regarding the operation of precision seeders with pneumatic distribution, which could be used as a new study model for optimizing these seeders, depending on the physical characteristics of the seeds from different crops.